# Fear Level Classification Based on Emotional Dimensions and Machine Learning Techniques

**DOI:** 10.3390/s19071738

**Published:** 2019-04-11

**Authors:** Oana Bălan, Gabriela Moise, Alin Moldoveanu, Marius Leordeanu, Florica Moldoveanu

**Affiliations:** 1Department of Computer Science and Engineering, Faculty of Automatic Control and Computers, University POLITEHNICA of Bucharest, 060042 Bucharest, Romania; alin.moldoveanu@cs.pub.ro (A.M.); marius.leordeanu@cs.pub.ro (M.L.); florica.moldoveanu@cs.pub.ro (F.M.); 2Department of Computer Science, Information Technology, Mathematics and Physics (ITIMF), Petroleum-Gas University of Ploiesti, 100680 Ploiesti, Romania; gmoise@upg-ploiesti.ro

**Keywords:** fear classification, emotional assessment, feature selection, affective computing

## Abstract

There has been steady progress in the field of affective computing over the last two decades that has integrated artificial intelligence techniques in the construction of computational models of emotion. Having, as a purpose, the development of a system for treating phobias that would automatically determine fear levels and adapt exposure intensity based on the user’s current affective state, we propose a comparative study between various machine and deep learning techniques (four deep neural network models, a stochastic configuration network, Support Vector Machine, Linear Discriminant Analysis, Random Forest and k-Nearest Neighbors), with and without feature selection, for recognizing and classifying fear levels based on the electroencephalogram (EEG) and peripheral data from the DEAP (Database for Emotion Analysis using Physiological signals) database. Fear was considered an emotion eliciting low valence, high arousal and low dominance. By dividing the ratings of valence/arousal/dominance emotion dimensions, we propose two paradigms for fear level estimation—the two-level (0—*no fear* and 1—*fear*) and the four-level (0—*no fear*, 1—*low fear*, 2—*medium fear*, 3—*high fear*) paradigms. Although all the methods provide good classification accuracies, the highest F scores have been obtained using the Random Forest Classifier—89.96% and 85.33% for the two-level and four-level fear evaluation modality.

## 1. Introduction

Emotion is defined as a conscious mental reaction subjectively experienced and directed towards a specific object, accompanied by physiological and behavioral changes in the body [1]. The field of affective computing aims to enhance the interaction between the human and the machines by identifying emotions and designing applications that automatically adapt to these changes. The term emerged in 1997, proposed by Rosalind Picard [2] and since then it has drawn the attention of researchers from interdisciplinary domains, being at the confluence of psychology, medicine and computer science. With applications in education, cognitive behavioral sciences, healthcare and entertainment, affective computing deals with recognizing and modelling human emotions in a way that would improve overall user experience. In order to achieve this goal, various mapping, thresholding and machine learning techniques have been employed for relevant feature selection and emotion classification. To classify emotions, some discrete and dimensional models have been proposed and applied throughout the years. The discrete models of affect rely on the existence of a set of basic emotions from which more complex ones derive. For instance, Plutchik [3] proposed a psycho-evolutionary theory built on eight basic emotions—anger, fear, sadness, disgust, surprise, anticipation, acceptance and joy. On the other hand, Ekman [4] envisioned six basic emotions—anger, disgust, fear, joy, sadness and surprise. The *dimensional models* are based on a multi-dimensional space where each axis represents the values of an emotional component. The bipolar model (known also as the Circumplex Model of Affect) [5], describes an emotion by taking into account the *valence* and *arousal* dimensions. Valence ranges from negative to positive, whereas arousal indicates how strong the emotion is perceived, ranking from not excited to excited. A third dimension is *dominance*, defined as the capacity of being in control of one’s emotions. 

This paper extracts and classifies the feeling of fear from the EEG and biophysical recordings stored in the DEAP database using different machine and deep learning algorithms [6]. We considered that a discrete emotion is the result of the combination of two or three emotional dimensions [7,8]. Therefore, *fear was described by negative valence, high arousal and low dominance* [9]. Our purpose is to define accurate methods of fear level classification, as we intend to develop a system for treating phobias based on gradual exposure to gamified scenarios. The system would automatically determine the user’s fear level, based on the EEG and physiological signals recorded in real-time and increase or decrease the exposure intensity according to the subject’s emotional changes. In order to achieve this, we propose two paradigms: one based on two fear levels and the other based on four fear levels.

The recordings from the DEAP database were divided into two and four groups, for each paradigm. One classification (the two-level fear evaluation modality) groups the data into a *no fear* cluster (0) and a *fear* class (1), while the other (the four-level fear evaluation modality) divides into *no fear* (0), *low fear* (1), *medium fear* (2) and *high fear* (3). The clustering was performed considering equal division (in two parts and four parts) of the valence, arousal and dominance intervals. The unsupervised K-Means algorithm validated our division approach with 87% accuracy and then the data was fed to four deep neural network models, a stochastic configuration network (SCN) [10,11] and four machine learning techniques (Support Vector Machine (SVM), Linear Discriminant Analysis (LDA), Random Forest (RF) and k-Nearest Neighbors (kNN)) for training and cross-validation. We tested our paradigms with five sets of input features based on physiological recordings and EEG data: (1) The 32-channel raw EEG values (as stored in the DEAP database) and the physiological recordings (hEOG, vEOG, zEMG, tEMG, Galvanic Skin Response (GSR), Respiration, Plethysmography (PPG) and temperature); (2) Power Spectral Density (PSD) of all 32 EEG channels in the alpha, beta and theta frequency ranges and the physiological recordings (hEOG, vEOG, zEMG, tEMG, GSR, Respiration, PPG and temperature); (3) Petrosian Fractal Dimensions of 32 EEG channels and the physiological recordings (hEOG, vEOG, zEMG, tEMG, GSR, Respiration, PPG and temperature); (4) Higuchi Fractal Dimension of 32 EEG channels and the physiological recordings (hEOG, vEOG, zEMG, tEMG, GSR, Respiration, PPG and temperature); (5) Approximate Entropy for each of the 32 EEG channels and the physiological recordings (hEOG, vEOG, zEMG, tEMG, GSR, Respiration, PPG and temperature). More comparative studies were performed for each condition in order to obtain the best fear classifier for each case. 

High classification accuracies were obtained with no feature selection, as well as when the Fisher, Principal Component Analysis (PCA) and (Sequential Feature Selector) SFS feature selection techniques were employed.

Our approach is original and of actual importance. As far as we know, no affective assessment research has been performed in which the emotional dimensions (valence, arousal and dominance) contribute to the formulation of fear levels. This is a relevant topic, as fear recognition and classification contribute to the development of applications treating prevailing medical and behavioral conditions, such as phobias or post-traumatic stress disorder.

## 2. Materials and Methods

### 2.1. Fear Description

Fear is definitely characterized by low valence (negative feeling) and high arousal. So is anger. However, what differentiates these two is the fact that in the case of anger, one dominates his feelings, whereas in the case of fear, the person loses control of his reactions, experiencing submission and passivism [9]. Fear starts in the presence of a stressful stimulus perceived by the sensory organs (eyes, ears, skin) and ends with the release of chemical substances generating bodily reactions such as high heart rate, fast breathing and muscular activation during the fight-or-flight response. The parts of the brain involved in fear processing are; the thalamus—which transmits sensory data; the sensory cortex—interprets sensory information; the hippocampus—decodes the stimuli in order to determine similar contexts and recalls related memories; the amygdala—almond-shaped set of nuclei in the temporal lobe, being at the same time the most sensitive part of the brain in response to fear; and the hypothalamus—which activates the fight-or-flight response through the sympathetic nervous and adrenal-cortical systems. Some researchers claim that fear is somehow processed differently than other emotions, by omitting the sensory cortex from the path to the amygdala. The temporal length of the trail is 12 milliseconds, a genetic evolutionary consequence that can make the difference between life and death in response to dangerous stimuli [12]. This theory clarifies the reasons why phobias and anxiety disorders are caused by unconscious reactions or fears that the person himself/herself is unable to understand or explain. To a certain extent, fear is healthy. It has supported survival and evolution throughout centuries and is an everyday component of our lives. However, irrational, prolonged fear is both emotionally and physically demanding, as it impedes people from successfully performing ordinary activities such as meeting others, walking in crowded spaces, speaking in public or exposing themselves to heights. This is the case of phobia, an anxiety disorder produced by a persistent and irrational fear, caused by past traumatic experiences implying exposure to fear-provoking stimuli. There are three types of phobias—agoraphobia (fear of public places), social phobia (fear of meeting or relating to other people) and specific phobias (anxieties produced by particular objects and situations). Phobias manifest themselves through intense activity in the nervous and sympathetic systems—increased heart rate, fast breathing, sweating, higher body temperature and other symptoms that can conduct to panic attacks and losing one’s consciousness. The modality of treating phobias is fear extinction, that works by creating a conditioned response reversing the fear response. The fear extinction memory resides in the medial prefrontal cortex and overrides the fear memory formed in the amygdala. This can be achieved by in-vivo and virtual exposure to fear-provoking stimuli in a controlled environment and Cognitive Behavioral Therapy (CBT), a healing treatment where the patients learn to acknowledge their fears and the causes producing them.

### 2.2. EEG and Biophysical Signals

The physiological responses are produced either by the Central Nervous System, consisting of the brain and the spinal cord and the Autonomic Nervous System that reflects cardiovascular, electrodermal and muscular changes. The cardiovascular activity is measured using electrocardiography (ECG), from which heart rate (HR) and heart rate variability (HRV) are calculated. The blood volume pulse is recorded by a photoplethysmography (PPG) device that comes in contact with the skin and measures the changes in cardiac electrical potential over time. The PPG device is usually attached to the finger or ear lobe, emits infrared light and records how much is being reflected back. Anxiety and fear produce an increase in HR of over 40 bpm from baseline [13]. While the normal values in resting state are between 60 and 100 bpm, when the subject experiences fear, his HR level can exceed 110 bpm [13]. The electrodermal activity (EDA) or galvanic skin response (GSR) represents a measure of the skin’s electrical conductivity and a reliable response to external stimuli. For measuring GSR, two electrodes are placed on the distal phalanges of the middle and index fingers. Conductivity is ensured by an increased amount of sweat (produced by the sweat glands) that has two components—a baseline tonic skin conductance and a phasic skin conductance that is highly responsive to excitation. While the baseline values are around 0.03 and 0.05 microSiemens, threatening stimuli produce a raise to around 2–3 microSiemens or extreme values of 8 microSiemens [14]. Besides, the mean value of GSR is correlated with arousal [15]. Other common physiological responses indicating emotional changes are electromyography (EMG) that records the electrical activity of the muscles, respiration rate and body temperature [16]. In [17], GSR and EMG were highly correlated with arousal and valence for all subjects. Irregular respiration and fast breathing are linked to aroused emotions like fear. Facial temperature significantly decreased and blinking rate increased after watching a horror movie in the fear assessment research presented in [18].

EEG is a technique responsible for recording and interpreting the brain’s electrical activity. The cortical activity is based on the connection and communication between neurons, made possible by the transmission of small electrical signals called electrical impulses. They create potentials that are received by the electrodes placed on the user’s scalp, and the pathways recorded by the EEG device represent the potential differences between selected electrodes. Generated by the flow of electrical stimulation between the 100 billion neurons, brainwaves are divided into different bandwidths, with specific functions and levels of complexity. The delta waves (0.5–3 Hz) are generated in deep sleep, being useful for healing and regenerating purposes. The theta waves (3–8 Hz) occur in dreams, imagination and deep meditation. The alpha waves (8–12 Hz) are the gateway to meditation, relaxation, open-eyes dreaming, learning, the feeling of being present. They originate in the occipital area and advance to the frontal sides of the brain that process emotion and behavior. High alpha intensities are correlated with relaxation, whereas the emergence of cognitive activities lead to a phenomenon called “alpha blocking,” defined by a decrease of alpha amplitudes. The beta waves (12–30 Hz) are responsible for states of attention, alertness, cognition, decision-making and problem-solving. The fastest brainwaves, the gamma (30–42 Hz), dominate our perception and consciousness, being strongly connected to the generation of positive, highly spiritual feelings and esoteric experiences.

Besides the valence scheme, the approach/withdrawal theory proposed by Davidson [19] relies on the motivational dimension of the emotional changes. Approach is associated with a positive feeling and is characterized by activation in the left frontal hemisphere of the brain, while withdrawal is mapped by a stimulation of the right cortical area. This corresponds to a decrease of alpha intensity in the left frontal hemisphere when positive emotions are elicited and an increase of alpha amplitudes in the right frontal brain areas whenever withdrawal processes take place [19,20,21,22]. However, this theory is not generally applicable, as other studies [23] showed that positive emotions are correlated with an increase of brain activity in the right area for four individuals. In what concerns the beta band, the higher the beta amplitudes, the higher is the anxiety. Both horror movies and virtual reality exposure to horror gameplay yielded an increase of 24 microvolts, respectively 33 microvolts in the range of the beta waves [13]. Valence is associated with theta frontal asymmetry [24] and gamma activation at the temporal lobes [25]. Valence was also positively correlated with left parietal-occipital power in the theta band and negatively correlated with right posterior alpha power. An increase in beta activation in the left-central area and a decrease in the right frontal cortex is mapped to high valence. As the gamma waves are concerned, there is a positive correlation between valence and right posterior gamma activation [26]. Moreover, frontal and fronto-medial regions have been activated in the theta range in the presence of positive stimuli [27] and in the delta frequency bands when negative stimulation has been provided [28]. The ratio of the slow waves to the fast waves (delta/beta or theta/beta) [29,30,31,32] has been extensively used for fear detection, showing that this ratio is negatively correlated with fear. In [18], it was also observed that the power ratio between the delta and beta bands decreases after watching a horror movie. In what concerns arousal, corresponds to a decrease of right posterior alpha power [26], an increase in the lower bands such as delta [28] and theta [33] and higher gamma amplitudes [34].

### 2.3. Machine Learning Techniques for Emotion Recognition

Various machine and deep learning technique have been used for feature selection and emotion classification in affective computing studies. Feature selection simplifies the model, reduces the training time and prevents overfitting. The most used feature selection algorithms are Principal Component Analysis (PCA), Sequential Forward Selection (SFS), ANOVA and Fisher linear discriminant. In combination with efficient algorithms such as Support Vector Machine (SVM), k- Nearest Neighbors (kNN), Linear Discriminant Analysis (LDA), Random Forest classifier, Naïve Bayes Classifier and artificial neural networks (both shallow and deep), a lot of studies have tried to construct solid emotion classification models, relying on a combination of dimensions from the Circumplex Model of Affect—valence, arousal, dominance, liking etc.

Having EEG input as features, [35] obtained an accuracy rate of 62% for recognizing pleasant, neutral and unpleasant emotions with Short Time Fourier Transform and the SVM classifier. Similarly, using the alpha frequencies and the SVM algorithm, a mean accuracy of 45% was obtained for the same classification pattern of pleasant, neutral and unpleasant feelings [36]. Using the FPz and the F3/F4 channels, a classification accuracy of 97% for arousal and 94% for valence was achieved in [37]. The division of power of the left to the right channels for 12 electrode pairs conducted to a classification success of 82% [38] and the use of the Higher Order Crossings in [39] led to an accuracy of 83%. Both studies employed the SVM technique. In [40], the Higuchi Fractal Dimension values from the FC6 electrode were used for arousal recognition and the difference between the AF3 and F4 electrodes for valence identification. The application worked in real-time and ensured recognition of six basic emotions—fear, frustration, sadness, happiness, pleasure and satisfaction. In [41], a Bayes classification framework was used for recognizing three affective classes—calm, positive excited and negative excited with an accuracy of 63.4%. Using a Long Short-Term Memory (LSTM) network and having as input raw EEG signals, the research in [42] succeeded to converge to an average binary classification accuracy of 85% for arousal and valence, respectively (low arousal/high arousal and low valence/high valence). Their results are better than those of; Koelstra et al. [6]—62% and 56%; Atkinson and Campos [43]—73% and 73%; Yoon and Chung [44]—70% and 70%; Naser and Saha [45]—66% and 64% for arousal and valence. Masood and Farooq [46] used Common Spatial Patterns (CSP) from 14 EEG channels and LDA to analyze fear emotions using two different stimuli and obtained classification accuracies between 55% and 73%.

Liu [47] presents a comparative study of four machine learning techniques for classifying anxiety, boredom, engagement, frustration and anger. SVM offered the best classification accuracy (85.81%), followed by Regression Tree (83.5%), kNN (75.16%) and Bayesian Network (74%). By performing feature selection, the classification of the last two methods increased by 4%. In [48] Sequential Floating Forward Search and Fisher Projection were used for classifying eight emotions with an accuracy rate of 81%. In [49,50], SVM was conducted to an accuracy rate of 78%, 61% and 41% for recognizing three, four and five affective classes. The approach proposed by Chanel [51] aimed at adapting a Tetris game’s difficulty levels according to the emotional output of the players. Without feature selection, there was an accuracy of 55% for peripheral features and 48% for EEG features (using the LDA algorithm, closely followed by SVM, with 47%). The Fast Correlation-Based Filter (FCBF) feature selection method increased the accuracy estimation of the peripheral features to 59% and ANOVA, to 56%, of the EEG features. After the fusion of EEG and peripheral signals, there was an increase of classification accuracy that reached 63%. The study described in [6], having as input the recordings from the DEAP database, conducted a classification accuracy of 62% (EEG) and 57% (peripheral signals) for arousal and 57% (EEG) and 62% (peripheral signals) for valence.

In a review on adaptation in affective video games, Bontchev [16] concludes that SFS is the most used feature selection technique and that LDA is widely employed for classifying emotions in small datasets, considering the Power Spectral Density values of the EEG frequencies. For larger datasets, SVM and artificial neural networks are more popular.

### 2.4. The DEAP Database

There are public emotion databases containing a large number of EEG and peripheral recordings that can be used for research purposes. Moreover, some of them have pre-processed versions available, well-suited to those who want to apply classification techniques without the inconvenience of processing the data first.

The DEAP [6] database contains physiological recordings (EEG and peripheral signals) and subjective ratings of 32 participants (50% male and 50% female, aged 19–37, mean age 26.9) who watched 40 one-minute long music videos. The users rated the videos in terms of arousal, valence, dominance, liking and familiarity. For 22 subjects, frontal face videos were also recorded. The experiment started with a 2 min baseline recording, in which the participants were asked to relax, while a fixation cross was displayed on the screen. Each trial consisted of a 5 s baseline recording (fixation cross), 1 min display of the music video and the self-assessment of arousal, valence, liking and dominance on a continuous 9-point scale using a Self-Assessment Mannequin. The subjects had to move the mouse cursor below the numbers and indicate their rating. The following EEG channels of the 10/20 system were recorded at a sampling rate of 512 Hz using AgCl electrodes: FP1, AF3, F3, F7, FC5, FC1, C3, T7, CP5, CP1, P3, P7, PO3, O1, Oz, Pz, FP2, AF4, Fz, F4, F8, FC6, FC2, Cz, C4, T8, CP6, CP2, P4, P8, CP6, CP2, P4, P8, PO4, O2. The peripheral signals were: GSR, respiration amplitude, skin temperature, electrocardiogram, blood volume, electrooculogram (EOG) and EMG of Zygomaticus and Trapezius muscles.

The processed DEAP dataset was made available at [52]. It contains 128-Hz down-sampled and segmented raw data in the form of 32 files, one file for each subject. Each file contains two arrays: a 40 × 40 × 8064 data array (video × channel × data) and a 40 × 4 labels array (video × label, where label represents the rating for valence, arousal, dominance and liking). The EEG data was processed as follows: the EOG artefacts were removed using a source separation technique, a bandpass frequency filter from 4 to 45 Hz was applied, the data was averaged to the common reference by subtracting the mean of the values and the 3-s pre-trial baseline was also removed. In addition, the peripheral physiological signals were down-sampled to 128 Hz and had their 3-s pre-trial baseline removed.

Other public databases are MAHNOB-HCI, containing 32-channel EEG and peripheral data recorded from 27 individuals who watched 20 video clips and pictures [53]. The peripheral signals were ECG, GSR, skin temperature and respiration volume. In addition, eye gaze and face videos have been captured and the users rated arousal, valence and dominance on a discrete scale from 1 to 9. The difference between DEAP and MAHNOB-HCI is that DEAP stores plethysmograph measurements, while MAHNOB-HCI contains recorded electrocardiograms. In the MAHNOB-HCI dataset there is no electromyogram data. The SEED database [54] contains raw EEG recordings from 15 subjects who watched Chinese videos. In their study, emotion assessment was performed by applying questionnaires immediately after watching each clip.

However, the most popular and consistent emotion database remains DEAP, as it has been used for research purposes in a multitude of studies. Containing recordings from a large number of individuals and having pre-processed data in both Matlab and Python formats, a comprehensive documentation and an appropriate description, we consider the DEAP database as a reliable solution for extracting and classifying specific emotions, in particular fear.

### 2.5. Our Paradigms for Fear Level Classification

Fear is described as a low valence, high arousal and low dominance emotion [9]. As the subjective ratings of valence, arousal and dominance from the DEAP database are continuous values between 1 and 9, we evaluated fear based on two modalities: the two-level and the four-level division of fear intensity. The two-level modality divides fear into *no fear* (0) and *fear* (1), while the four-level modality divides into *no fear* (0), *low fear* (1), *medium fear* (2) and *high fear* (3). Table 1 and Table 2 present the division of the continuous (1–9) values of valence, arousal and dominance for these two evaluation modalities, into two and four levels of fear. The length of the intervals is four for the two-level condition and two for the four-level condition. Thus, for the two-level evaluation modality, *fear* (1) is characterized by valence in the interval [1; 5], arousal [5; 9] and dominance [1, 5), while *no fear* (0) is described by valence in the interval (5; 9], arousal [1; 5) and dominance [5; 9]. We used these divisions to label the corresponding data from DEAP.

Figure 1 presents the steps to get to the two classifiers. After extracting and labeling the data, we validated our labeling using K-Means. Five input features sets were generated for each paradigm: (1) 32-channel raw EEG values (as stored in the DEAP database) and the physiological recordings (hEOG, vEOG, zEMG, tEMG, GSR, Respiration, PPG and temperature); (2) Power Spectral Density (PSD) of all 32 EEG channels in the alpha, beta and theta frequency ranges and the physiological recordings (hEOG, vEOG, zEMG, tEMG, GSR, Respiration, PPG and temperature); (3) Petrosian Fractal Dimensions of 32 EEG channels and the physiological recordings (hEOG, vEOG, zEMG, tEMG, GSR, Respiration, PPG and temperature); (4) Higuchi Fractal Dimension of 32 EEG channels and the physiological recordings (hEOG, vEOG, zEMG, tEMG, GSR, Respiration, PPG and temperature); (5) Approximate Entropy for each of the 32 EEG channels and the physiological recordings (hEOG, vEOG, zEMG, tEMG, GSR, Respiration, PPG and temperature). More Machine Learning (ML) and feature selection techniques were applied for both the two-level and four-level evaluation modalities.

### 2.6. K-Means Clustering

In order to validate these evaluation modalities, the unsupervised K-Means algorithm was applied on the ratings from the dataset—each of the 32 subjects has 40 ratings, one for each video he/she watched. The K-Means technique is a popular clustering method that divides unlabeled observations into classes (or clusters) with the nearest mean, called cluster centroid. By using the Python implementation from the scikit-learn library [55], we obtained a prediction accuracy of 87% for the two-level fear evaluation modality. In short, 87% of the ratings evaluated by the two-level modality as belonging to the category *fear/no fear* were classified in the same cluster by the K-Means technique. This result strongly supports the psychological inference that fear is characterized by low valence, high arousal and low dominance. There is a total number of 198 entries for the *no fear* condition and 174 for the *fear* condition in the two-level evaluation modality. For the four-level modality, we have seven entries for *no fear*, 60 for *low fear*, 42 for *medium fear* and 35 for *high fear*. As the EEG and physiological recording had a duration of 60 s, in order to enlarge the training dataset, we divided each 60 s long recording into 12 5-s long segments. Thus, we finally obtained 4464 entries for the two-level fear evaluation modality and 1728 entries for the four-level modality to be provided to the classifiers for training and cross-validation.

### 2.7. Fear Level Classification

For classifying the fear ratings, we applied machine and deep learning algorithms and feature selection techniques for both the two-level and four-level evaluation modalities. The features were the EEG (raw values/Power Spectral Densities in the alpha, beta and theta frequency ranges/Approximate Entropy/Petrosian Fractal Dimension/Higuchi Fractal Dimension), hEOG (horizontal EOG), vEOG (vertical EOG), zEMG (Zygomaticus EMG), tEMG (Trapezius EMG), GSR, Respiration, PPG and temperature. The deep learning methods applied were four deep neural networks (DNN1-DNN4) with different numbers of hidden layers and neurons per layer, SVM, RF, LDA and kNN and the feature selection techniques were: Fisher selection, PCA and SFS.

DNN1 is composed of one input layer, three hidden layers and one output layer, where each hidden layer has a number of 300 neurons per layer. For the two-level scale, we used the binary crossentropy loss function in the output layer (with two possible outputs, 0 or 1, where 0 stands for *no fear* and 1 for *fear*), while for the four-level scale, the categorical crossentropy loss function was employed with one-hot encoding that creates four output values (0–3), one for each class (*no fear, low fear, medium fear, high fear*). The model uses the Adam gradient descent optimization algorithm and the Rectified Linear Unit (RELU) activation function on each layer. The data was previously standardized to reduce it to zero mean and unit variance, and the Keras classifier [56] received 1000 epochs for training and a batch size of 20. The cross-validation was performed by using the k-fold method with k = 10 splits.

DNN2 has three hidden layers and 150 neurons per layer, DNN3 has six hidden layers and 300 neurons per layer and DNN4 has six hidden layers with 150 neurons per layer. Stochastic Configuration Networks (SCNs) randomly assign the input weights and biases of the hidden nodes in a supervised way. Deep SCNs can be designed more easily than classical deep neural networks and present a higher consistency between learning and generalization, offering a fast and efficient path to problem solving [57]. The Matlab code is available on the DeepSCN website [58]. For training and testing the SCN, the DEAP dataset was divided into 70% training and 30% test, with a training tolerance of 0.01, using a cross-validation procedure repeated 10 times.

For the SVM (with rbf kernel), RF, LDA and kNN algorithms, the data was divided into 70% training and 30% test, using the train_test_split method from the sklearn library. Each classification method was trained and cross-validated 10 times, without feature selection and with the Fisher, PCA and SFS feature selection methods respectively. The average accuracies and F1 scores of these 10 iterations were calculated for each feature selection modality in part.

## 3. Results

First, we considered as input the 32-channel raw EEG values (as stored in the DEAP database) and the physiological recordings (hEOG, vEOG, zEMG, tEMG, GSR, Respiration, PPG and temperature). In this case, the DNN models receive, at input, 40 neurons, corresponding to the 40 features (32 raw EEGs + eight peripherals). The cross-validation results obtained after performing training and testing using the machine and deep learning methods previously described, with and without feature selection, are presented in Table 3. For the DNN models, feature selection is not necessary, as we used the dropout regularization technique that prevents overfitting.

Secondly, we calculated the Power Spectral Density (PSD) of all 32 EEG channels in the alpha, beta and theta frequency ranges using the OpenVibe [59] software. Then, we computed the average alpha, beta and theta PSD values in the pre-frontal (FP), AF (between FP and F), frontal (F), FC (between F and C), central (C), temporal (T), P (parietal), CP (between C and P), O (occipital) and PO (between P and O) sides of the brain. This resulted in 30 EEG inputs, which, added to the eight peripheral values, reach to a number of 38 input features fed to the classifiers. The cross-validation results obtained after performing training and testing using the machine and deep learning methods, with and without feature selection, are presented in Table 4.

Then, we computed the Petrosian Fractal Dimension, the Higuchi Fractal Dimension and Approximate Entropy for each of the 32 EEG channels using the functions implemented in the PyEEG [60] library. Combined with the eight peripheral features, we reached a total number of 40 features provided as input to the classifiers. The cross-validation results obtained after performing training and testing using the machine and deep learning methods, with and without feature selection, are presented in Table 5, Table 6 and Table 7.

## 4. Discussion

### 4.1. Classification Accuracy

For the condition where the raw EEG values were included in the model, for the two-level evaluation modality, the highest F scores were obtained using kNN (86.81% with no feature selection, 84.70% using Fisher selection and 85.54% with PCA) and SVM (60%, using SFS selection). For the four-level evaluation modality, the highest F scores were obtained using RF (83.55% with no feature selection), kNN (79.22% using the Fisher selection technique and 81.67% using PCA) and SVM and LDA, both scoring an accuracy of 48% when the SFS algorithm was employed. In the case of the condition where the alpha, beta and theta amplitudes were included in the model, for the two-level evaluation modality, the highest F scores were obtained using kNN (85.84% with no feature selection and 87.45% with PCA) and RF (85.68%, using Fisher selection and 81% using SFS). For the four-level evaluation modality, the highest F scores were obtained using RF (85.33% with no feature selection and 68% with SFS selection) and kNN (82.60% using the Fisher selection technique and 82.58% using PCA). For the condition where the Petrosian fractal dimensions were included in the model, for the two-level evaluation modality, the highest F scores were obtained using RF (89.67% with no feature selection, 87.33% using Fisher selection and 80% using SFS selection) and kNN (85.37%, with PCA). For the four-level evaluation modality, the highest F scores were obtained using kNN (84.83% with no feature selection, 77.09% with Fisher and 83.50% with PCA feature selection) and RF (66%, by employing the SFS selection algorithm). In the case of the condition where the Higuchi fractal dimensions were included in the model, for the two-level evaluation modality, the highest F scores were obtained using RF (89.96% with no feature selection, 88.96% with Fisher selection and 81% using SFS) and kNN (84.31% with PCA). For the four-level evaluation modality, the highest F scores were obtained using RF (82.59% with no feature selection, 77.32% with Fisher and 68% with SFS) and kNN (81.01% using PCA feature selection). Lastly, for the condition where the Approximate Entropies were included in the model, for the two-level evaluation modality, the highest F scores were obtained using RF (89.65% with no feature selection, 89.516% with Fisher selection and 80% using SFS) and kNN (84.93% with PCA). For the four-level evaluation modality, the highest F scores were obtained using RF (80.78% with no feature selection, 80.47% with Fisher and 63% with SFS) and kNN (75.97% using PCA feature selection). All these maximum F scores are highlighted in Table 3, Table 4, Table 5, Table 6 and Table 7.

### 4.2. Selected Features

The Fisher feature selection algorithm ranked as most relevant features the raw EEG values of the F4, FC2, CP5 and C3 electrodes, the alpha amplitude in the AF channel, the beta amplitudes in AF and PO, the theta intensities of the AF, P, O and PO electrodes, the Pz Petrosian value, FC5 and C3 for Higuchi, PO3 and O1 Approximate Entropies. The most selected peripheral features were PPG, temperature and respiration rate. The SFS algorithm selected the FP1 and AF3 raw EEGs, the beta amplitudes in the AF and F channels and the theta amplitudes in the C and CP channels respectively, the FC1 and CP1 Petrosian values, the F3, CP1 and P3 Higuchi values and the FP1, AF3 and F7 Approximate Entropies. The highest ranked peripheral features were GSR, tEMG and respiration rate.

### 4.3. Cortical Activation

For both the two-level and four-level fear evaluation modalities, we computed the differences in the alpha and theta frequency bands between the right and the left frontal sides of the brain (the FP, AF and F electrodes), the mean amplitude of the beta waves in the left central and right frontal cortical areas, as well as the ratio of slow waves to fast waves (theta/beta). For the two-level fear evaluation modality, we applied the Mann–Whitney non-parametrical statistical significance test between the samples corresponding to the *no fear*/*fear* conditions. The null hypothesis assumed that there is no difference between the distributions of the data samples. Table 8 presents the *p* values and the means of the two samples. For all the cases, the *p* value was lower than the alpha threshold set at 0.05 and therefore the null hypothesis was rejected, indicating that the samples come from different distributions.

The results obtained for the alpha and theta asymmetry evaluation cases indicate that the difference between the right and the left frontal amplitudes in these two frequency bands is lower when the user does not experience fear (−0.12) than when he feels it (−0.05). However, the difference is negative, suggesting that there is less activation (less alpha and theta power) in the right cortical area than in the left one. Moreover, fear is associated with higher beta activation in the left central and right frontal parts of the brain (3.41 and 3.51) and a higher ratio of slow waves to fast waves (1.16).

For the four-level evaluation modality, we used the Kruskal–Wallis H test for multiple samples, where the null hypothesis states that all sample distributions are equal. For all our cases, the *p* value was lower than the alpha threshold set at 0.05 and therefore the null hypothesis was rejected. Table 9 presents the *p* values and the means of the four sample conditions with: *no fear* (0), *low fear* (1), *medium fear* (2) and *high fear* (3).

The results obtained for the alpha and theta asymmetry evaluation cases indicate that there is less activation in the right cortical area than in the left one (negative difference between these two). The difference increases with fear (higher for the *medium fear* and *high fear* condition) than for the *no fear* and *low fear* conditions. The high fear situation conducted the highest activation in the left central (3.54) and right frontal (3.79) cortical sides for the beta waves. Moreover, the ratio of theta to beta waves increases with fear (1.24 and 1.18 for the *medium* and *high fear* conditions, compared to 1.04 and 1.17 for the user experiences *low fear* or *no fear* at all).

## 5. Conclusions

This paper presented an exhaustive approach towards extracting and classifying the feeling of fear from the EEG and peripheral physiological recordings stored in the DEAP database using advanced machine and deep learning techniques. The unsupervised K-Means clustering algorithm grouped the recordings into two and four clusters, corresponding to the two-level and four-level fear evaluation modalities. The two-level modality groups the entries into two categories: *no fear* and *fear*, while the four-level evaluation modality groups them into *no fear*, *low fear*, *medium fear* and *high fear*. With an accuracy of 87%, K-Means validated the psychological inference that fear is characterized by low valence, high arousal and low dominance. For the two-level evaluation modality, the highest F scores were obtained by using the Random Forest Classifier—89.96% accuracy with no feature selection and Higuchi Fractal Dimension, 89.51% with Fisher selection and approximate entropy, 81% using SFS feature selection with both EEG Power Spectral Density and Higuchi Fractal Dimension. kNN conducted the highest classification accuracy (87.45%) with the PCA feature selection and Power Spectral Densities as input. In addition, for the four-level fear evaluation modality, the highest F scores were obtained also using the Random Forest Classifier—85.33% with no feature selection and PSD, 68% with SFS for PSD and Higuchi Fractal Dimensions. With kNN, the highest accuracy was obtained using Fisher selection with PSD (82.6%) and PCA feature selection with Petrosian Fractal Dimension (83.5%).

Our classification results are comparable or higher than the best results from the literature, which were obtained by Lin et al. [38] (82%), Petrantonakis et al. [39] (83%) using Higher Order Crossings and the SVM technique and Alhagry et al. [42] (85%), who trained and tested a Long-Short Term Memory network.

The study of Lin et al. [38] involved the participation of 26 subjects whose EEG data was recorded from 32 channels while listening to emotion-eliciting music stimuli. They targeted four basic emotional—anger, joy, sadness and pleasure—separated into different categories of valence and arousal: anger—negative valence and high arousal; joy—positive valence and high arousal; pleasure—positive valence and low arousal; sadness—negative valence and low arousal. Classification was performed using the SVM classifier with RBF kernel. Emotion classification yielded an accuracy of 82%, while classifying valence and arousal into low/high levels, 86%, respectively 85%.

Petrantonakis et al. [39] recorded EEG signals (three EEG channels, Fp1, Fp2, and a bipolar channel of F3 and F4) from 16 subjects who looked at pictures of facial affect expressing six basic emotions—happiness, surprise, anger, fear, disgust and sadness. The subjects were asked to comprehend the emotion presented and then mark each picture according to the emotion he considers it conveys. Higher Order Crossings (HOC) was used for feature extraction and QDA, kNN, SVM and Mahalanobis distance used for classification. The EEG data was examined using both single-channel and combined-channels scenarios. The highest classification accuracies were 63% (using QDA) for single channel and 83% (using SVM) for combined channels.

Alhagry et al. [42] also used the DEAP database to classify the raw EEG signals into low/high arousal, valence and liking. This binary classification was done using a Long-Short Term Memory (LSTM) network which consisted of two fully connected LSTM layers, a dropout layer that prevents overfitting and a dense layer responsible for classification. The model was trained on 75% of the data using four-fold cross-validation and tested on 25% of them. The classification accuracies were 85.65%, 85.45%, and 87.99% for arousal, valence and liking.

Our method defines a complex emotion such as fear as a combination of valence, arousal and dominance (low valence, high arousal and low dominance, supported by the results obtained by clustering using the K-Mean algorithm with and accuracy of 87%) and classifies it on a two-level and four-level scale. Thus, we did not perform just a binary classification for dividing the data into low or high valence, arousal and dominance, but a much complex approach was proposed, that of defining an emotion as a combination of emotional dimensions and classifying it into using two scales: the two-level (0—*no fear* and 1—*fear*) and the four-level (0—*no fear*, 1—*low fear*, 2—*medium* fear, 3—high fear) scale. Moreover, we used as the feature set the entire set of data from the DEAP database, which includes both EEG and physiological signals. Alhagry et al. [42], on the contrary, classified using only the raw EEG signals. Our approach is stronger as we combined various classification and feature extraction techniques with various modalities of analyzing and decomposing the EEG data—raw, PSD, Petrosian Fractal Dimensions, Higuchi Fractal Dimension and Approximate Entropy—which all provided notable results.

A positive correlation between the fear level intensity and the difference between the Power Spectral Densities in the right and left frontal cortical areas in the alpha and theta range was observed, although this difference is negative, meaning that there is more alpha and theta power in the left side of the brain. Intensity of the left central and right frontal beta increases with fear. Moreover, there is a positive correlation between fear and the ratio of slow to fast waves, for both fear evaluation modalities.

The integration of deep and machine learning techniques into the development of emotion recognition systems improves their reliability and efficiency. Fear classification is a topic of actual importance, given the prevalence of the affective computing domain. Therefore, its automatic assessment can be integrated into the development of therapeutic applications for treating behavioral conditions such as phobias and post-traumatic stress disorder or for adapting the exposure intensity in a gamified environment where the system acts as a virtual therapist [61,62]. As future research ideas, we intend to run the same feature selection and classification algorithms on the recordings from the MAHNOB database and other available datasets, as well as to develop a system for treating phobias based on gradual exposure to virtual reality, that automatically determines fear levels from EEG and biophysical data.

## Figures and Tables

**Figure 1 sensors-19-01738-f001:**
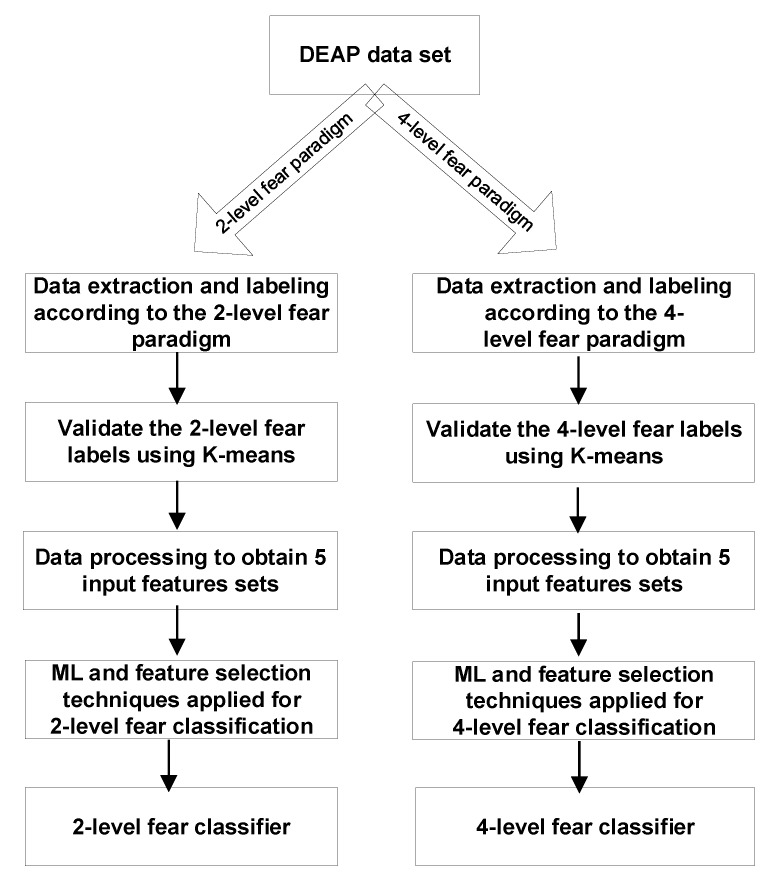
Steps for obtaining the two classifiers.

**Table 1 sensors-19-01738-t001:** Division of valence, arousal and dominance for the two-level fear evaluation modality.

Label	Valence	Arousal	Dominance
*No fear* (0)	(5; 9]	[1; 5)	[5; 9]
*Fear* (1)	[1; 5]	[5; 9]	[1; 5)

**Table 2 sensors-19-01738-t002:** Division of valence, arousal and dominance for the four-level fear evaluation modality.

Label	Valence	Arousal	Dominance
*No fear* (0)	[7; 9]	[1; 3)	[7; 9]
*Low fear* (1)	[5; 7)	[3; 5)	[5; 7)
*Medium fear* (2)	[3; 5)	[5; 7)	[3; 5)
*High fear* (3)	[1; 3)	[7; 9]	[1; 3)

**Table 3 sensors-19-01738-t003:** Classification accuracy when input is a vector of 32 raw electroencephalograms (EEGs) and eight peripheral signals. (The highest accuracies are written with bold.)

Type of Feature Selection	Classifier	Fear Evaluation Modality			
		Two-Level		Four-Level	
		F1 Score (%)	Accuracy (%)	F1 Score (%)	Accuracy (%)
No feature selection	DNN1	70.59	70.95	58.78	59.84
	DNN2	67	67.34	34.16	45.78
	DNN3	71.91	71.95	47.69	51.16
	DNN4	69.17	69.27	24.51	41.67
	SCN	75.15	76	48.35	49
	SVM	73.5	74	64.65	66.09
	RF	85.63	86	**83.85**	**84.01**
	LDA	60.19	61	55.77	56.65
	kNN	**86.81**	**87**	82.52	82.66
Fisher	SVM	69.95	70.90	60.03	62.24
	RF	84.41	84.55	77.81	78.03
	LDA	53.29	55.90	46.46	48.17
	kNN	**84.70**	**84.70**	**79.22**	**79.19**
PCA	SVM	74.20	74.87	69.93	70.83
	RF	82.30	82.45	78.32	78.54
	LDA	56.87	58.27	46.43	49.36
	kNN	**85.54**	**85.53**	**81.67**	**81.75**
SFS	SVM	**60**	**60**	**48**	**48**
	RF	57	57	43	43
	LDA	59	59	**48**	**48**
	kNN	57	57	46	46

**Table 4 sensors-19-01738-t004:** Classification accuracy when input is a vector of 30 alpha, beta and theta PSDs and eight peripheral signals. (The highest accuracies are written with bold.)

Type of Feature Selection	Classifier	Fear Evaluation Modality			
		Two-Level		Four-Level	
		F1 Score (%)	Accuracy (%)	F1 Score (%)	Accuracy (%)
No feature selection	DNN1	81.99	81.99	67.46	68.98
	DNN2	78.16	78.14	55.92	58.85
	DNN3	82.21	82.26	57.70	60.94
	DNN4	79.14	79.12	30.13	43.63
	SCN	75.12	75.5	51.2	51.5
	SVM	83.15	83.13	83.46	84.01
	RF	93.11	93.13	**85.33**	**85.74**
	LDA	70.46	70.52	60.98	61.46
	kNN	**85.84**	**85.82**	82.94	83.24
Fisher	SVM	78.15	78.13	76.79	77.07
	RF	**85.68**	**85.75**	80.28	80.54
	LDA	64.90	65.37	51.66	52.99
	kNN	81.05	81.04	**82.60**	**82.66**
PCA	SVM	85.82	85.81	82.52	82.85
	RF	84.75	84.83	81.08	81.27
	LDA	65.94	66.13	54.93	55.39
	kNN	**87.45**	**87.44**	**82.58**	**82.77**
SFS	SVM	71	71	56	56
	RF	**81**	**81**	**68**	**68**
	LDA	64	64	48	48
	kNN	78	78	61	61

**Table 5 sensors-19-01738-t005:** Classification accuracy when input is a vector of 32 Petrosian Fractal Dimensions and eight peripheral signals. (The highest accuracies are written with bold.)

Type of Feature Selection	Classifier	Fear Evaluation Modality			
		Two-Level		Four-Level	
		F1 Score (%)	Accuracy (%)	F1 Score (%)	Accuracy (%)
No feature selection	DNN1	80.90	80.91	62.65	64.35
	DNN2	77.65	77.64	39.56	48.96
	DNN3	80.08	80.17	49.11	56.60
	DNN4	76.47	76.50	24.51	41.67
	SCN	78.6	78.75	47.34	48.15
	SVM	81.57	81.57	82.01	82.47
	RF	**89.67**	**89.78**	83.05	83.62
	LDA	64.02	64.03	66.90	67.44
	kNN	84.79	84.78	**84.83**	**84.97**
Fisher	SVM	81.49	81.49	72.96	73.60
	RF	**87.33**	**87.54**	68.57	69.75
	LDA	62.94	62.99	52.09	52.79
	kNN	83.23	83.21	**77.09**	**77.26**
PCA	SVM	81.77	81.78	79.05	79.69
	RF	79.35	79.62	70.68	71.46
	LDA	64.41	64.58	63.98	64.32
	kNN	**85.37**	**85.36**	**83.50**	**83.64**
SFS	SVM	71	71	56	56
	RF	**80**	**80**	**66**	**66**
	LDA	67	67	51	51
	kNN	78	78	61	61

**Table 6 sensors-19-01738-t006:** Classification accuracy when input is a vector of 32 Higuchi Fractal Dimensions and eight peripheral signals. (The highest accuracies are written with bold.)

Type of Feature Selection	Classifier	Fear Evaluation Modality			
		Two-Level		Four-Level	
		F1 Score (%)	Accuracy (%)	F1 Score (%)	Accuracy (%)
No feature selection	DNN1	81.40	81.41	59.89	62.67
	DNN2	77.01	76.99	36.96	47.74
	DNN3	81.09	81.14	49.11	57.12
	DNN4	78.51	78.52	24.51	41.67
	SCN	77.15	78.5	45.25	46.20
	SVM	81.64	81.64	80.85	81.70
	RF	**89.96**	**90.07**	**82.59**	**83.24**
	LDA	69.09	69.10	64.96	65.32
	kNN	83.38	83.36	80.52	80.73
Fisher	SVM	80.75	80.75	71.58	72.45
	RF	**88.96**	**89.10**	**77.32**	**72.83**
	LDA	66.59	66.87	55.35	56.45
	kNN	83	82.99	75.61	75.92
PCA	SVM	82.16	82.16	77.79	78.63
	RF	82.26	82.41	78.55	78.90
	LDA	69.06	69.16	61.38	61.89
	kNN	**84.31**	**84.30**	**81.01**	**81.21**
SFS	SVM	74	74	58	58
	RF	**81**	**81**	**68**	**68**
	LDA	66	66	48	48
	kNN	78	78	61	61

**Table 7 sensors-19-01738-t007:** Classification accuracy when input is a vector of 32 Approximate Entropies and eight peripheral signals. (The highest accuracies are written with bold.)

Type of Feature Selection	Classifier	Fear Evaluation Modality			
		Two-Level		Four-Level	
		F1 Score (%)	Accuracy (%)	F1 Score (%)	Accuracy (%)
No feature selection	DNN1	79.95	80.17	57.96	61.86
	DNN2	79.02	79.21	48.20	54.34
	DNN3	80.12	80.58	52.05	59.95
	DNN4	79.96	80.40	27.55	41.90
	SCN	80.20	80.40	51.25	51.30
	SVM	74.71	74.70	57.85	62.43
	RF	**89.65**	**89.78**	**80.78**	**81.70**
	LDA	62.81	62.91	49.12	52.22
	kNN	84.54	84.55	71.15	71.87
Fisher	SVM	75.75	75.75	60	64.35
	RF	**89.51**	**89.63**	**80.47**	**81.50**
	LDA	58.46	59.93	46.09	50.67
	kNN	84.96	85	78.82	79.38
PCA	SVM	77.63	77.78	64.63	68
	RF	82.40	82.54	73.49	74.03
	LDA	62.75	63.05	48.80	52.45
	kNN	**84.93**	**84.95**	**75.97**	**76.61**
SFS	SVM	72	72	55	55
	RF	**80**	**80**	**63**	**63**
	LDA	64	64	48	48
	kNN	78	78	62	62

**Table 8 sensors-19-01738-t008:** *p* values and means for the two-level fear evaluation condition during the Mann–Whitney test.

Evaluation Case	*p* Value	Mean *No Fear* Condition	Mean *Fear* Condition
Alpha frontal asymmetry	1.66 × 10^−10^	−0.12	−0.05
Left central beta	6 × 10^−3^	3.31	3.41
Right frontal beta	5.38 × 10^−28^	3.23	3.51
Theta frontal asymmetry	8.4 × 10^−11^	−0.21	−0.12
Ratio theta/beta	7.09 × 10^−8^	1.11	1.16

**Table 9 sensors-19-01738-t009:** *p* values and means for the four-level fear evaluation condition during the Kruskal–Wallis test.

Evaluation Case	*p* Value	Mean *No Fear* Condition	Mean *Low Fear* Condition	Mean *Medium Fear* Condition	Mean *High Fear* Condition
Alpha frontal asymmetry	3.85 × 10^−8^	−0.12	−0.20	−0.04	−0.08
Left central beta	2.84 × 10^−6^	3.25	3.32	3.22	3.54
Right frontal beta	3.43 × 10^−25^	2.92	3.27	3.19	3.79
Theta frontal asymmetry	6.01 × 10^−6^	−0.36	−0.25	−0.13	−0.18
Ratio theta/beta	3.4 × 10^−9^	1.04	1.17	1.24	1.18

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
