# Peer review of "Fear Level Classification Based on Emotional Dimensions and Machine Learning Techniques"

_sensors, 2019, doi:10.3390/s19071738_

Round 1
Reviewer 1 Report
This paper compares some ML techniques for a specific application (fear recognition), which is useful to online learning or other associated fields. Overall, the submission is well-written and organized, technical contributions are not obvious but it is sufficient to support an application type of publication in Sensors.
RF is a class of randomized method in ML, recent development of randomized learning techniques (S. Scardapane and D. Wang, Randomness in neural networks: an overview, WIREs Data Mining Knowledge Discovery, 2017, 7:e1200. doi: 10.1002/widm.1200; D. Wang and M. Li, Stochastic configuration networks: fundamentals and algorithms, IEEE Trans. On Cybernetics, vol. 47, no. 10, 3466-3479, 2017.) should be mentioned in this work. It will be interesting to see some results from stochastic configuration networks (advanced randomized learning, code can be found at www.deepscn.com) in the tables. If the authors cannot implement SCN-based classification system for this dataset, this method should be recommended in your future studies.
Author Response
Dear reviewer,
Thank you for the suggestion of using Stochastic Configuration Networks. They are a really interesting approach and it has been a pleasure to discover them.
Thus, even if it was in a very short time, we managed to adapt our procedure and train & test using the code you provided at http://deepscn.com/. We “translated” our approach in Matlab (with the other machine and deep learning techniques we worked with Python) and obtained some interesting results. We run the network 10 times in a cross-validation procedure (70% of the data has been randomly assigned for training and 30% randomly assigned for testing) and kept the same parameter settings as you provided in your examples. At the end of this file you can find two files with the Matlab code, for training and testing with both the 2-level and 4-level fear evaluation modalities.
The results we obtained have been added in Tables 3-7. Moreover, we cited 4 of the references you suggested, namely:
10. Scardapane, S. and Wang, D. Randomness in neural networks: an overview. WIREs Data Mining Knowledge Discovery. 2017, 7:e1200. doi: 10.1002/widm.1200
11. Wang, D. and Li, M. Stochastic configuration networks: fundamentals and algorithms. IEEE Trans. On Cybernetics. 2017, 47, 10, 3466-3479
57. Wang, D. and Li, M., Deep stochastic configuration networks with universal approximation property. Proceedings of 2018 International Joint Conference on Neural Networks, July 8-13, 2018, Rio de Janeiro, Brazil
58. Deep Stochastic Configuration Networks. Available online: http://www.deepscn.com/index.php (accessed on 21 March 2019)
In order to improve the Conclusions section, we made a comparison with other studies and added the following paragraph:
“Our classification results are comparable or higher than the best results from the literature, which have been obtained by Lin et al [38] (82%), Petrantonakis et al [39] (83%) using Higher Order Crossings and the SVM technique and Alhagry et al [42] (85%), who trained and tested a Long-Short Term Memory network. However, their methods classified emotions as low/high valence or low/high arousal. Our method detaches from the state-of-the-art, as it defines a complex emotion such as fear as a combination of valence, arousal and dominance and classifies it on a 2-level and 4-level scale.”
We hope you will find our manuscript suitable for publication in the Sensors journal and look forward to hearing from you.

Reviewer 2 Report
The paper is well written and well structured.
I have only one concern, that I think must be clarified: why are the authors assuming that the fear distribution is uniform? Why the cluster the obtain are of the same size, an thus the classification task balanced? I do not agree with this approach, and I think it should be revised by the authors...
Author Response
Dear reviewer,
From the DEAP dataset, for the 2-level fear evaluation modality, we extracted 198 records corresponding to no fear (labelled with 0) and 174 records corresponding to fear (labelled with 1). They have been put together and, in a 10-fold cross-validation procedure, divided into training and test dataset. The classifiers learned on the training dataset, a model has been constructed and consequently tested on the test dataset. For the 4-level fear evaluation procedure, we extracted 7 records for the no fear condition (labelled with 0), 60 for low fear (labelled with 1), 42 for medium fear (labelled with 2) and 35 for high fear (labelled with 3). The same procedure of training and testing has been performed as well. Thus, the clusters have different dimensions and the classification is not balanced.
If you refer to the division of fear levels (2-level and 4-level), we obtained them based on the emotional dimensions of valence, arousal and dominance, according to the intervals presented in Table 1 and Table 2. Of course, fear is a very complex emotion, but, for research purposes and considering the psychological theory of affect, we divided it into 2 or 4 levels, assuming it is a combination of low valence, high arousal and low dominance subjectively rated in the DEAP database by the people participating in their experiment on a scale from 1 to 9.
In order to improve our paper, as suggested by Reviewer 1, we added another classification technique, namely Stochastic Configuration Network. The results are presented in Tables 3-7. In order to strengthen the importance of our results, we made a comparison with other studies and added the following paragraph:
“Our classification results are comparable or higher than the best results from the literature, which have been obtained by Lin et al [38] (82%), Petrantonakis et al [39] (83%) using Higher Order Crossings and the SVM technique and Alhagry et al [42] (85%), who trained and tested a Long-Short Term Memory network. However, their methods classified emotions as low/high valence or low/high arousal. Our method detaches from the state-of-the-art, as it defines a complex emotion such as fear as a combination of valence, arousal and dominance and classifies it on a 2-level and 4-level scale.”
We hope you will find our manuscript suitable for publication in the Sensors journal and look forward to hearing from you.

Round 2
Reviewer 2 Report
I accept the paper in its present status.
Author Response
Dear Academic Editor,
Here is the response which we also included in the text of the paper:
Our classification results are comparable or higher than the best results from the literature, which have been obtained by Lin et al [38] (82%), Petrantonakis et al [39] (83%) using Higher Order Crossings and the SVM technique and Alhagry et al [42] (85%), who trained and tested a Long-Short Term Memory network.
The study of Lin et [38] al involved the participation of 26 subjects whose EEG data have been recorded from 32 channels while listening to emotion-eliciting music stimuli. They targeted four basic emotional states - anger, joy, sadness and pleasure, separated into different categories of valence and arousal: anger - negative valence and high arousal, joy - positive valence and high arousal, pleasure - positive valence and low arousal, sadness - negative valence and low arousal. Classification has been performed using the SVM classifier with RBF kernel. Emotion classification yielded an accuracy of 82%, while classifying valence and arousal into low/high levels, 86%, respectively 85%.
Petrantonakis et al [39] recorded EEG signals (3 EEG channels, Fp1, Fp2, and a bipolar channel of F3 and F4) from 16 subjects who looked at pictures of facial affect expressing six basic emotions - happiness, surprise, anger, fear, disgust and sadness. The subjects were asked to comprehend the emotion presented and then mark each picture according to the emotion he considers it conveys. Higher Order Crossings (HOC) was used for feature extraction and QDA, kNN, SVM and Mahalanobis distance used for classification. The EEG data was examined using both single-channel and combined-channels scenarios. The highest classification accuracies have been of 63% (using QDA) for single channel and 83% (using SVM) for combined channels.
Alhagry et al [42] also used the DEAP database to classify the raw EEG signals into low/high arousal, valence and liking. This binary classification has been done using a Long-Short Term Memory (LSTM) network which consisted of two fully connected LSTM layers, a dropout layer that prevents overfitting and a dense layer responsible with classification. The model was trained on 75% of the data using 4-fold cross-validation and tested on 25% of them. The classification accuracies were 85.65%, 85.45%, and 87.99% for arousal, valence and liking.
Our method defines a complex emotion such as fear as a combination of valence, arousal and dominance (low valence, high arousal and low dominance, supported by the results obtained by clustering using the K-Mean algorithm with and accuracy of 87%) and classifies it on a 2-level and 4-level scale. Thus, we did not perform just a binary classification for dividing the data into low or high valence, arousal and dominance, but a much complex approach was proposed, that of defining an emotion as a combination of emotional dimensions and classifying it into using 2 scales: the 2-level (0 – no fear and 1 - fear) and the 4-level (0 – no fear, 1 – low fear, 2 – medium fear, 3 – high fear) scale. Moreover, we used as feature set the entire set of data from the DEAP database, which includes both EEG and physiological signals. Alhagry et al [42], on the contrary, classified using only the raw EEG signals. Our approach is stronger as we combined various classification and feature extraction techniques with various modalities of analyzing and decomposing the EEG data - raw, PSD, Petrosian Fractal Dimensions, Higuchi Fractal Dimension and Approximate Entropy, which all provided notable results.
Thank you for your suggestion of improving the article and hope you will find it suitable for publication.
All the best,
The authors
